# Electrothermal bipolar vessel sealing devices are associated with lower rates of postoperative complications compared to ultrasonic devices in vulvar cancer surgery

Milla K. Mörsky [1,2]*, Ilkka S. Kaartinen[2,3], Reita H. Nyberg[1,2]

1 Department of Gynecology and Obstetrics, Tampere University Hospital, The Wellbeing Services County of Pirkanmaa, Tampere, Finland, 2 Faculty of Medicine and Health Technology, Tampere University, Tampere, Finland, 3 Department of Musculoskeletal Surgery and Diseases, Tampere University Hospital, The Wellbeing Services County of Pirkanmaa, Tampere, Finland

* milla.morsky@tuni.fi

## Abstract

### Introduction

Electrothermal bipolar vessel sealing devices (EBVS) and ultrasonic devices (US) – collectively known as advanced hemostasis devices (AHDs) – are considered equally feasible in laparoscopic procedures. However, US devices have been demonstrated to be more susceptible to abnormal heat accumulation when activation cycles are rapidly repeated, causing results from laparoscopic procedures to be poorly translated to vulvar cancer surgery. In this study, we aimed to determine whether EBVS and US are comparable in terms of peri- and postoperative morbidity in vulvar cancer surgery.

### Methods

This retrospective single-center study comprised patients who underwent a primary vulvectomy, partial vulvectomy, or radical local resection with an AHD in Tampere University Hospital, Finland, in 2011–2023. Our primary outcome measure was the Clavien-Dindo grade, which measures the incidence and severity of postoperative complications in the early (30-day) postoperative period. Secondary outcome measures were blood loss, postoperative blood transfusions, operative time, the total volume of groin drain output, and length of hospital stay.

### Results

Eighty-six patients were included (EBVS n = 45, US n = 41). Postoperative complications (Clavien-Dindo grades II – V) were significantly less common in the EBVS group compared to the US group (60% vs 85% in the EBVS and US groups,

**Data availability statement:** Data cannot be shared publicly under current Finnish legislation governing the secondary use of social and health data (the Act on the Secondary Use of Health and Social Data). Data are available from the Finnish Social and Health Data Permit Authority (Findata) for researchers who meet the criteria for access to confidential data. Data permits for the data underlying the results presented in the study can be applied for from Findata's website: https://findata.fi/en/.

**Funding:** This research was in part supported by grants from the Finnish Society for Research in Gynecology & Obstetrics (MM), the Finnish Society for Gynecological Surgery (MM), Tampere University (MM), and the Competitive State Research Financing of the Expert Responsibility area of the Tampere University Hospital (grant 9AB108) (IK). Funder websites: the Finnish Society for Research in Gynecology & Obstetrics - https://gynekologiyhdistys.fi/synnytys-ja-naistentautiopin-tutkimussaatio/; the Finnish Society for Gynecological Surgery - https://www.gks.fi; Tampere University - https://www.tuni.fi/; the Competitive State Research Financing of the Expert Responsibility area of the Tampere University Hospital - https://www.pirha.fi/ammattilaiselle/tutkimus/tutkimusrahoitus/valtion-tutkimusrahoitus. The funders had no role in study design, data collection and analysis, decision to publish, or preparation of the manuscript.

**Competing interests:** The authors have declared that no competing interests exist.

**Abbreviations:** SLNB, sentinel lymph node biopsy; IFL, inguinofemoral lymphadenectomy; EBVS, electrothermal bipolar vessel sealing device; US, ultrasonic device; AHD, advanced hemostasis device; FIGO, Fédération Internationale de Gynécologie et d'Obstétrique; TNM, tumor, nodes, metastases; ASA [classification], American Society of Anesthesiologists [classification]; aOR, adjusted odds ratio; CI, confidence interval

respectively; p = 0.015). The difference was driven by a discrepancy in grade II complications (49% vs 71%), which consisted primarily of infections in both groups. In a multivariable regression analysis adjusting for the extent of surgery, the use of an EBVS device was independently associated with a lower likelihood of postoperative complications compared to US (aOR 0.3, 95%CI 0.1–0.9 for EBVS vs US; p = 0.030). Both the amount of operative blood loss (median (IQR) 50 (45–200) ml vs 150 (88–400) ml; p = 0.005) and length of hospital stay (median (interquartile range) 6 (4–8) vs. 8 (6–10) days; p = 0.002) were lower in the EBVS group, but surgical device did not independently predict the highest quartile of either variable. The amount of postoperative blood transfusions, operative time, or groin drain output did not significantly differ between the groups.

## Conclusions

The data from this study suggests electrothermal bipolar vessel sealing devices could reduce early postoperative complications, especially those related to the surgical site, in vulvar cancer surgery compared to ultrasonic devices. Prospective studies are needed to ensure the generalizability of the results.

## Introduction

The incidence of vulvar cancer has been steadily growing, especially in women <60 years in high-income countries [1]. Despite advances in both minimally invasive and reconstructive techniques, surgical treatment of vulvar cancer remains complicated by significant morbidity [2–4]. Short-term complication rates of 21.8%, 39.6%, and 54.2% have recently been reported for vulvar surgery only, vulvar surgery + sentinel lymph node biopsy (SLNB), and vulvar surgery + inguinofemoral lymphadenectomy (IFL), respectively [4]. While inguinofemoral lymphadenectomy remains the most significant risk factor for postoperative complications, surgical nodal assessment can seldom be avoided entirely due to the prognostic significance of nodal involvement [5,6]. Other perioperative practices thus need to be critically evaluated to reduce surgical morbidity.

Electrothermal bipolar vessel sealing devices (EBVS) and ultrasonic devices (US) – collectively known as advanced hemostasis devices (AHDs) – have been widely used in vulvar cancer surgery since the early 2010s. As both instrument types have demonstrated comparable perioperative outcomes in several laparoscopic and minor open surgeries [7–12], the choice of instrument is often dictated by either availability or the surgeon's preference. However, the risk for abnormal heat accumulation and lateral thermal spread in AHDs increases if activation cycles are rapidly repeated or prolonged; as a result, findings from studies on laparoscopic or minor open surgeries may be poorly translated to surgeries involving large-volume soft tissue resections [13,14]. Only a small number of studies have investigated AHDs and postoperative morbidity in the context of major soft tissue surgeries, and while both AHD subtypes have been suggested to reduce morbidity compared to traditional electrocautery

in such settings [15–23], no studies to date have compared EBVS to US in vulvar cancer surgery or similar extensive soft-tissue resections. Exploring this issue in a vulvar cancer patient population is critical, considering the high complication rate of vulvar cancer surgery and the distinct thermal profiles and hemostatic properties of the AHDs.

In this study, we aimed to compare the incidence and severity of early postoperative complications in patients having gone through curative vulvar cancer surgery with an EBVS device or a US device. Additionally, we aimed to compare operative blood loss, the need for postoperative blood transfusions, operative time, groin drain output, and length of hospital stay between the groups.

## Methods

### Study design and outcomes

We conducted a retrospective cohort study with a population that comprised patients having undergone a complete or partial vulvectomy, or a radical local resection, with an AHD in Tampere University Hospital, Finland. Inclusion criteria were a preoperatively confirmed squamous cell histology, surgeries with curative intent, and complete electronic operative records detailing the instruments used. Exclusion criteria were neoadjuvant chemotherapy, tumors of non-squamous histology, surgeries performed with palliative intent, and any history of previous vulvar surgery and/or radiotherapy.

Our primary outcome measure was Clavien-Dindo grade for the early (30-day) postoperative period. Secondary outcome measures were blood loss, postoperative blood transfusions, operative time, the total volume of groin drain output, and length of hospital stay.

### Data extraction

We reviewed the electronic operative records of all patients with vulvar cancer having undergone a complete vulvectomy, partial vulvectomy, or a radical local resection in our center in the year 2005 or after; an electronic operative record (Centricity™ Opera, GE Healthcare) was then introduced, making detailed data on the devices and supplies used in each surgery available. From this cohort, we identified patients who were operated on using an AHD (EBVS or US). Patients meeting the inclusion criteria were identified from 2011 to 2023, with US being utilized between the years 2011–2019 and EBVS between 2013–2023. All surgeons who operated during this time period utilized each instrument at some point and were familiar with the devices from laparoscopic surgeries. The choice of instrument was based on either availability or surgeon's preference.

Data on patient characteristics, all surgical procedures, and the postoperative course of each patient were collected from electronic health records, which were most recently accessed on April 8, 2025. Both FIGO and TNM stages were defined for each patient according to the 2021 FIGO classification. ASA classification was either extracted from the electronic health records (69.8%) or, if not available, retrospectively determined based on the preoperative description of the patient's physical status (30.2%). Data on antibiotic prophylaxis, surgical instruments, operative time, and total blood loss were extracted from electronic operative records.

Operative time was defined as the time from the first incision to wound closure. For patients having undergone a lymphadenectomy, a total volume of groin drain output was calculated for each groin. If data on the drain output was incomplete (i.e., the patient was discharged with a drain), the groin in question was excluded from the analysis regarding groin drain output.

### Clavien-Dindo grading

An uneventful course for postoperative recovery (i.e., Clavien-Dindo 0) was first defined to assess a Clavien-Dindo grade for each patient. A description of the classification system and examples of its application to this study population are presented in Table 1 [24]. Elective surgeries performed during the 30-day postoperative period were not classified as Clavien-Dindo grade III complications.

**Table 1. The Clavien-Dindo classification for early (30-day) postoperative course with examples of complications for each grade.**

| | | Definition | Examples |
|---|---|---|---|
| Grade 0 | | Normal postoperative course | Defined in this study as follows:<br>**Pain management**: Epidural analgesia for 1–3 days, followed by oxycodone-naloxone for 1–2 days, after which transitioning to non-opioid analgesics only<br>**Antiemetics**: permitted for up to 1 day postoperatively<br>**Antibiotics**: allowed only if a urethral resection was performed, with use restricted to the period until the urinary catheter is removed (approximately 3–5 days postoperatively)<br>**Laxatives**: permitted following perianal or rectal resection if deemed necessary by the surgeon |
| Grade I | | Any deviation from the normal postoperative course without the need for pharmacological treatment* or surgical, endoscopic and radiological interventions Infections opened bedside | IV fluidsImaging tests without interventionsDeviations from the normal postoperative course in the use of analgetics, antipyretics or antiemet-icsWound complications requiring conservative treatment only (dehis-cence, seroma, mild infections) |
| Grade II | | Complications requiring pharmacological treatment with drugs other than such allowed for grade I compli-cationsBlood transfusionsParenteral nutrition | Infections requiring antibiotic treatment (not included: prophylactic use described in grade 0)Pharmacological treatments not classified as grade 0 or grade IBlood transfusionsParenteral nutrition |
| Grade III | | Requiring surgical, endoscopic or radiological intervention | |
| | Grade IIIa | Intervention not under general anesthesia | Re-suturing or revision of the wound (necrosis, dehiscence, bleeding) Vacuum-assisted closure treatmentThoracocentesis |
| | Grade IIIb | Intervention under general anesthesia | Reoperations requiring fecal diversion |
| Grade IV | | Life-threatening complications (including stroke) requiring intermediate or intensive care | |
| | Grade IVa | Single-organ dysfunction (including dialysis) | StrokeHeart failure |
| | Grade IVb | Multi-organ dysfunction | |
| Grade V | | Death of a patient | |

*allowed therapeutic regimens are drugs as antiemetics, antipyretics, analgetics, diuretics, electrolytes, and physiotherapy.

## Surgical techniques

The extent of surgery was tailored according to the clinical characteristics of the primary tumor (size, multifocality, depth of invasion, and localization), and to imaging studies assessing the extent of disease. All patients received a single-dose prophylactic antibiotic before the first incision and a follow-up dose if operative time exceeded three hours. Low-molecular-weight heparin as thromboprophylaxis was started one day preoperatively and continued up to 30 days postoperatively. Incisions in the vulva and groins were performed with a monopolar blade, and the remaining steps were performed with an AHD. Vulvar resection was performed using lateral tumor-free margins of 1–2 cm, depending on the size and depth of invasion of the primary tumor, and the vulvar tissue was removed in all its thickness to the underlying urogenital diaphragm or perineal membrane. Lymphadenectomies were always performed from incisions separate to the vulvar resection. Unilateral groin procedures were deemed acceptable in case of lateral primary tumors (>1 cm from the midline). SLNBs were performed by identifying nodes with both technetium and blue dye and sending them for frozen section examination. A complete inguinofemoral lymphadenectomy was performed in case of a positive sentinel node. All groin drains were passive and removed when the drain output was < 30ml/day or as per the surgeon's instructions.

All reconstructions were performed by a plastic surgeon. In all cases, a local fasciocutaneous flap was chosen [VY advancement flap 13/14 (92.9%) vs 20/22 (90.9%) in the US and EBVS groups, respectively; propeller flap 1/14 (7.1%) vs 1/22 (4.5%); transposition flap 0/14 (0.0%) vs 1/22 (4.5%)]. The flaps were mobilized with monopolar diathermy.

## Statistical analysis

The patients were divided into two groups according to the energy source of the instrument used in their operation – US [SonoSurg™ (Olympus, Tokyo, Japan)] or EBVS [Voyant™ (Applied Medical, CA, USA), LigaSure™ (Medtronic, MN, USA), Thunderbeat™ (Olympus, Tokyo, Japan)]. Surgeries in which Thunderbeat™ was utilized (n = 1) were included in the EBVS group considering the device's primary energy source and vessel sealing properties.

Statistical tests were chosen based on the distribution of each variable, and the assumptions of each test were assessed prior to analysis. Continuous variables with a normal distribution were compared using Student's $t$ test, and are presented as mean ± standard deviation (SD). Non-normally distributed continuous variables were compared using Mann-Whitney $U$ test, and are presented as median (interquartile range).

Clavien-Dindo grades were crosstabulated and compared using Fisher's exact test. A multivariable binary logistic regression analysis was then performed for postoperative complications (Clavien-Dindo II – V). Odds ratios for surgical devices were adjusted for the type of vulvar surgery (complete radical vulvectomy or partial vulvectomy/radical local excision), type of groin surgery (IFL with or without SLNB, SLNB only, or none), reconstructions, and tumors either four centimeters or greater (≥ 4 cm) in diameter or TNM tumor category ≥T2. The variables were tested for multicollinearity (r < |0.5|).

Blood loss, operative time, and the length of hospital stay were compared using either Student's $t$ test or Mann-Whitney U test, depending on the distribution of the variable. If a statistically significant difference between the groups was found, a multivariable binary logistic regression analysis was performed with the > 75th percentile as the dependent variable: for blood loss, the dependent variable was > 300 ml, and for the length of hospital stay, > 9 days. Independent variables included in the model were identical to the ones included in the regression analysis for Clavien-Dindo II – V.

Groin drain output was compared with a Mann-Whitney U test in a per-groin analysis comprising all groins where an IFL was performed (48 and 25 groins in the US and EBVS groups, respectively).

All independent variables included in the regression analyses were chosen based on a literature search and careful clinical consideration and thus included in the final model. Adjusted odd ratios and their p-values were presented. All p-values are two-tailed, and a p-value of <0.05 was considered statistically significant.

All statistical analyses were performed using SPSS Statistics version 29.0.0.0 (IBM Corp., Armonk, NY, USA).

## Ethical considerations

This study was conducted under research permits granted by Tampere University Hospital Institutional Review Board (R16582, approved 27.10.2020) and the Finnish Social and Health Data Permit Authority Findata (THL/822/14.02.00/2022 and THL/270/14.06.00/2024; approved 17.5.2023 and 23.1.2024). Written informed consent from study participants was not obtained, as the Finnish legislation allows the secondary use of health and social data for research purposes under conditions described in the Act on the Secondary Use of Social and Health Data (552/2019), provided that a research permit has been approved by the National Health and Social Data Permit Authority (Findata). In accordance with the requirements of the Act, the data were pseudonymized after extraction and transferred to a secure operating environment audited by Findata.

The results are presented in accordance with the Finnish Social and Health Data Permit Authority Findata's guidelines on result anonymity; to protect the data subjects' anonymity, the minimum frequency presented in tables is ≤ 3 [25]. Additionally, if a variable contains only one cell with a censored value, the cell with the second-lowest value is presented as a range to ensure anonymity. Findata was responsible for ensuring the anonymity of the final results. Due to legal restrictions supporting data is not available (see "Data availability statement").

## Results

### Patient characteristics

86 patients met the inclusion criteria (Fig 1). Patient characteristics are presented in Table 2. No statistically significant difference was found in the patients' age, smoking status, body mass index, ASA classification, FIGO stage, TNM tumor

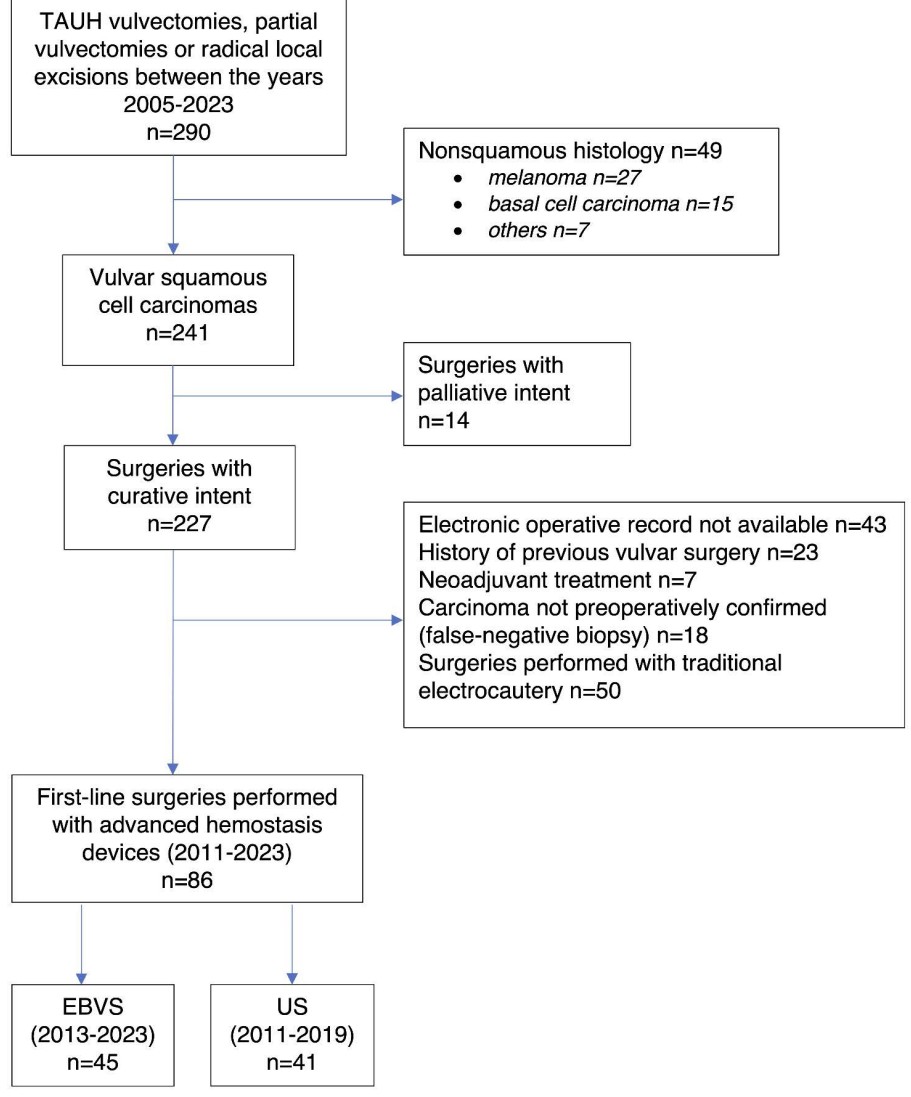

**Fig 1. Patient selection protocol.**

classification, frequency of reconstructions, or the size of the primary tumor between the groups. Surgeries in the EBVS group were slightly less extensive than in the US group (partial vulvectomies/radical local excisions 61.0% vs. 86.7%; p = 0.007, and SLNBs without lymphadenectomies 22.0% vs 48.9% in the US and EBVS groups, respectively; p = 0.030). The number of bilateral groin procedures was comparable between the groups. (Table 2)

### Clavien-Dindo grade

Clavien-Dindo II – V complications were more frequent in the US than in the EBVS group (85.4% vs 60.0% in the US and EBVS groups, respectively; p = 0.015). The most significant difference was found in the frequency of Clavien-Dindo II complications (70.7% vs 48.9%), of which infections accounted for a majority. (Table 3). After adjusting for the extent of surgery in a multivariable regression analysis, the use of an EBVS device was independently and significantly associated with a lower probability of Clavien-Dindo II – V complications compared to US (aOR 0.3, 95% CI 0.1–0.9; p = 0.030). Detailed results of the regression analysis are presented in Table 4.

**Table 2. Patient characteristics.**

| | | US (n = 41) | EBVS (n = 45) | P-value |
|---|---|---|---|---|
| Age, years (mean ± SD) | | 73.0 ± 10.5 | 71.9 ± 8.5 | 0.60 |
| BMI, kg/m2 (median (interquartile range)) | | 27.4 (25.7–32.3) | 27.5 (25.2–32.0) | 0.90 |
| Smoking, n (%) | | | | 0.75 |
| | Yes | 4 (9.8%) | 7 (15.6%) | |
| | No | 30 (73.2%) | 31 (68.9%) | |
| | Missing | 7 (17.1%) | 7 (15.6%) | |
| ASA classification, n (%) | | | | 0.40 |
| | I | 4–6 (9.8–14.6%) | ≤3 (≤6.7%) | |
| | II | 16 (39.0%) | 20 (44.4%) | |
| | III | 18 (43.9%) | 22 (48.9%) | |
| | IV | ≤3 (≤7.3%) | ≤3 (≤6.7%) | |
| Primary tumor (T) TNM category, n (%) | | | | 0.30 |
| | TIa | 3–5 (7.3–12.1%) | ≤3 (≤6.7%) | |
| | TIb | 36 (87.8%) | 38 (84.4%) | |
| | TII | ≤3 (≤7.3%) | 4–6 (8.9–13.3%) | |
| | TIII | 0 (0.0%) | 0 (0.0%) | |
| FIGO stage, n (%) | | | | 0.93 |
| | I | 24 (58.5%) | 27 (60.0%) | |
| | II | ≤3 (≤7.3%) | ≤3 (≤6.7%) | |
| | III | 15–17 (36.6–41.5%) | 15–17 (33.3–37.8%) | |
| | IV | 0 (0.0%) | 0 (0.0%) | |
| Primary tumor size, mm (median (interquartile range)) | | 20.0 (9.0–41.0) | 28.0 (19.0–35.0) | 0.27 |
| Type of vulvar surgery, n (%) | | | | 0.007* |
| | Radical vulvectomy | 16 (39.0%) | 6 (13.3%) | |
| | Partial vulvectomy or radical local excision | 25 (61.0%) | 39 (86.7%) | |
| Reconstruction, n (%) | | 14 (34.1%) | 22 (48.9%) | 0.19 |
| Groin procedures, n (%) | | | | 0.030* |
| | None | 4 (9.8%) | ≤3 (≤6.7%) | |
| | SLN biopsy only | 9 (22.0%) | 22 (48.9%) | |
| | IFL | 28 (68.3%) | 20–22 (44.4–48.9%) | |
| Bilateral groin procedures | | 28 (68.3%) | 35 (77.8%) | 0.34 |
| Inguinal node metastasis, n (%) | | 16 (39.0%) | 16 (35.6%) | 0.83 |

Cells with low frequencies are presented in accordance with the Finnish Social and Health Data Permit Authority (Findata) requirements for result anonymity.

### Blood loss, operative time, groin drain output, and length of hospital stay

The amount of operative blood loss was lower in the EBVS group (median 150 ml (IQR 87.5–400) vs 50 ml (IQR 45–200); p = 0.005) (Table 3), but after adjusting for the extent of surgery in a multivariable regression analysis, only the extent of vulvar surgery was an independently significant predictor of blood loss exceeding the 75th percentile, i.e., over 300 ml (aOR 4.0, 95% CI 1.1–14.8 for radical vulvectomy vs partial vulvectomy/radical local excision; p = 0.032) (Table 5). The number of postoperative blood transfusions did not significantly differ between the groups. (Table 3) Similarly, the length of hospital stay was significantly shorter in the EBVS group (median 6 (IQR 4–8) vs 8 (IQR 6–10) days; p = 0.002) (Table 3), but the surgical device did not independently predict a hospital stay of over 9 days (>75th percentile) (Table 6). There was no significant difference in operative time or total volume of groin drain output between the groups (Table 3).

**Table 3. Clavien-Dindo grade, blood loss, operative time and groin drainage by surgical instrument used.**

| | | US (n = 41) | EBVS (n = 45) | P-value |
|---|---|---|---|---|
| Clavien-Dindo grade, n (%) | | | | 0.027* |
| | 0 | 0 (0%) | 5 (11.1%) | |
| | I | 6 (14.6%) | 13 (28.9%) | |
| | II | 29 (70.7%) | 22 (48.9%) | |
| | IIIa | ≤3 (≤7.3%) | ≤3 (≤6.7%) | |
| | IIIb | ≤3 (≤7.3%) | ≤3 (≤6.7%) | |
| | IVa | ≤3 (≤7.3%) | 0 (0%) | |
| | IVb | ≤3 (≤7.3%) | 0 (0%) | |
| | V | 0 (0%) | 0 (0%) | |
| Clavien-Dindo II – V, n (%) | | 35 (85.4%) | 27 (60.0%) | 0.015* |
| | Infections | 28 (68.3%) | 25 (55.6%) | 0.23 |
| | Postoperative blood transfusions | 4 (9.8%) | ≤3 (≤6.7%) | 0.19 |
| Dehiscence | | 16 (40.0%) | 25 (56.8%) | 0.12 |
| Blood loss, ml (median (interquartile range)) | | 150 (87.5–400) | 50 (45–200) | 0.005* |
| Operative time, min (mean ± SD) | | 160.8 ± 68.3 | 171.8 ± 73.3 | 0.48 |
| Length of hospital stay, d (median (interquartile range)) | | 8 (6–10) | 6 (4–8) | 0.002* |
| **Per groin analysis (n = 73)** | | **US (n = 48)** | **EBVS (n = 25)** | **P-value** |
| Total volume of groin drain output, ml (median (interquartile range)) | | 255.0 (138.8–383.8) | 215 (130.0–377.5) | 0.39 |

Cells with low frequencies are presented in accordance with the Finnish Social and Health Data Permit Authority (Findata) requirements for result anonymity.

**Table 4. Predictors of immediate postoperative complications. Multivariable binary logistic regression for Clavien-Dindo ≥ II complications. (n = 86).**

| Factor | | Adjusted OR (95% CI) | P-value |
|---|---|---|---|
| Reconstruction | | 1.0 (0.3–2.9) | 0.92 |
| Groin procedures | | | 0.44 |
| | IFL (vs. SLNB only) | 2.0 (0.6–6.9) | 0.25 |
| | None (vs. SLNB only) | 0.8 (0.1–5.8) | 0.35 |
| Radical vulvectomy (vs. partial vulvectomy or radical local excision) | | 0.8 (0.2–3.3) | 0.72 |
| Tumor ≥ 4 cm or TNM ≥ T2 | | 2.0 (0.5–8.1) | 0.36 |
| Surgical instrument (EBVS vs. US) | | 0.3 (0.1–0.9) | 0.030* |

**Table 5. Predictors of blood loss > 300 ml (> 75th percentile). Multivariable binary logistic regression. (n = 86).**

| Factor | | Adjusted OR (95% CI) | P-value |
|---|---|---|---|
| Reconstruction | | 2.8 (0.9–9.4) | 0.09 |
| Groin procedures | | | 0.80 |
| | IFL (vs. SLNB only) | 0.6 (0.2–2.5) | 0.50 |
| | None (vs. SLNB only) | 0.8 (0.1–9.8) | 0.89 |
| Radical vulvectomy (vs. partial vulvectomy or radical local excision) | | 4.0 (1.1–14.8) | 0.032* |
| Tumor ≥ 4 cm or TNM ≥ T2 | | 3.4 (0.9–12.9) | 0.07 |
| Surgical instrument (EBVS vs. US) | | 0.3 (0.1–1.1) | 0.08 |

**Table 6. Predictors of prolonged hospital stay (> 75th percentile, i.e., >9 days). Multivariable binary logistic regression. (n = 86).**

| Factor | | Adjusted OR (95% CI) | P-value |
|---|---|---|---|
| Reconstruction | | 2.1 (0.6–7.5) | 0.27 |
| Groin procedures | | | 0.55 |
| | IFL (vs. SNB only) | 2.6 (0.5–15.4) | 0.28 |
| | None (vs. SNB only) | 2.5 (0.2–37.5) | 0.50 |
| Radical vulvectomy (vs. hemivulvectomy or radical local excision) | | 0.7 (0.2–3.1) | 0.72 |
| Tumor ≥ 4 cm or TNM ≥ T2 | | 1.9 (0.5–7.5) | 0.36 |
| Surgical instrument (EBVS vs. US) | | 0.3 (0.1–1.3) | 0.12 |

## Discussion

### Summary of main results

In this retrospective cohort study, we found that the use of electrothermal bipolar vessel sealing devices was independently associated with fewer early postoperative complications following vulvar cancer surgery compared to ultrasonic devices. The lower number of overall complications in the EBVS group was primarily driven by a lower incidence of Clavien-Dindo II complications, particularly infections. EBVS was associated with a lower amount of operative blood loss and a shorter hospital stay; however, the type of surgical device did not independently predict neither blood loss nor hospital stay exceeding the highest quartile (>300ml or >9 days, respectively), nor was there a statistically significant difference in the frequency of postoperative blood transfusions. We found no difference in operative time, or the duration or volume of lymph drain output between the two device groups.

### Results in the context of published literature

Surgical morbidity remains a crucial issue in the treatment of vulvar cancer patients, and there is active ongoing research aiming to reduce the radicality of the procedure without compromising oncologic safety [26–30]. In recent years, advanced hemostasis devices – especially EBVS devices – have been suggested to reduce peri- and postoperative morbidity compared to traditional electrocautery in various types of soft-tissue surgeries; however, few studies have compared surgical instruments in vulvar cancer surgery [15–21,23]. Pellegrino et al. were the first to report a reduction in both operative time and blood loss in vulvectomy + inguinofemoral lymphadenectomy with US compared to traditional electrocautery, but the study is from an era before EBVS devices [31]. A more recent randomized controlled trial by Pouwer et al. found that EBVS devices were associated with reduced short-term morbidity compared to traditional electrocautery after inguinofemoral lymphadenectomy [22]. However, whether there are differences in operative outcomes between the AHD subtypes (i.e., EBVS and US) in vulvar cancer surgery has not been previously studied. A possible difference in postoperative outcome is thus a novel finding and bears clinical relevance, particularly as knowledge on the operative outcomes of AHD subtypes is limited and derived mainly from studies focusing on laparoscopic or minor open surgeries.

In this study, Clavien-Dindo II was the most frequently occurring class of complications: 70.7% vs 48.9% of all patients in the US and EBVS groups, respectively, were classified as Clavien-Dindo II. This group of complications consisted primarily of infections: postoperative blood transfusions were rare in both groups (9.8% vs ≤ 6.7%; p = 0.19), and no cases of parenteral nutrition were observed. However, the higher percentage of radical vulvectomies in the US group must be considered when interpreting the results, as both the risk for complications and operative blood loss are closely linked to the extent of the surgery [4,32]. Nevertheless, EBVS was independently associated with a lower likelihood of postoperative complications in a multivariable analysis adjusting for the extent of surgery. This finding suggests EBVS and US devices could have clinically relevant differences in one or more properties that have the potential to impact postoperative outcomes, which, in electrosurgery, are likely related to thermal damage, hemostatic control, or operative time.

In electrosurgery, the risk for lateral thermal spread and subsequent thermal damage is closely linked to the number and duration of the activation cycles [13,33]. Both EBVS and US produce minimal lateral thermal spread with optimal use and a single-bite setting [33,34], but US devices are shown to be more susceptible to abnormal heat accumulation and increased thermal spread if activation is either prolonged or rapidly repeated, particularly if tissue bites are thin or partial [13,14]. Shibao et al. demonstrated the temperature in the jaws and blades of the US device to exceed 300C with such usage; subsequently, cooling times of up to 1.5 minutes were needed for the tip to reach a safe temperature [13]. This could pose a risk for thermal injury when using an US device in vulvar cancer surgery – multiple consecutive tissue bites may be needed when performing resections of large surface areas, and the low need for dissection and ease of maneuvering in an open-surgery setting cause a lack of spontaneous breaks between activation cycles that would facilitate the cool-down of the device. Furthermore, the more volatile thermal profile of the US device could make it harder for the surgeon to estimate whether the cutting blade has reached a safe temperature. A lower likelihood of complications – particularly infections – in the EBVS group could thus be related to a lower amount of thermal damage a more favorable tissue response; however, this topic needs to be explored in a prospective study.

Our results suggest EBVS and US devices provide a comparable hemostasis in vulvar cancer surgery. Although the US group had higher amounts of operative blood loss, the difference seems to be driven by the higher number of radical vulvectomies in the US group as the extent of vulvar surgery was the only independently significant predictor of major blood loss in the multivariable regression model. Moreover, the similar hemostatic properties of EBVS and US devices is supported by the comparable amount of postoperative blood transfusions between the groups. While it is difficult to establish a cut-off value for a clinically relevant amount of operative blood loss, both anemia and blood transfusions are known risk factors for postoperative complications [24,35–37]. Most previously published studies comparing EBVS and US in an open surgery setting have concluded that the devices' hemostatic properties are comparable, although these studies have mainly focused on surgical interventions where the expected amount of blood loss is relatively small and, arguably, less impactful on postoperative outcomes [9,10,17].

We found no significant difference in operative time between the EBVS and US groups; this finding is in line with several previously published studies [9,17,38]. Thus, the observed difference in the frequency of postoperative complications in the EBVS and US groups seems to not be explained by operative time. Similarly, the total volume of groin drain output did not differ between the groups, consistent with prior studies on inguinal or axillary lymphadenectomies [17,22]. EBVS has been suggested to be associated with a higher incidence of lymphoceles following axillary lymphadenectomies [17], but in the present study, lymphoceles could not be included as a secondary outcome due to a lack of systematic data on their presence at the time of the postoperative follow-up visit. Nevertheless, this study question remains highly relevant due to the aforementioned differences in thermal profile between EBVS and US devices and is hopefully explored in future studies.

The length of hospital stay was shorter in the EBVS group compared to the US group, which is likely associated to the reduced number of postoperative complications in the EBVS group. A historical bias must, however, be considered when interpreting the results, as ongoing efforts to streamline postoperative management have likely led to shorter hospital stays during the EBVS era compared to the US era. This may also be reflected in the more favorable postoperative outcomes of the EBVS group. Our results thus encourage exploring this topic in a randomized, prospective setting, as the question of whether the risk for thermal injury in vulvar cancer surgery might be more easily mitigated with an EBVS device should be raised.

## Strengths and weaknesses

This study has several limitations and strengths. The main limitation of this study is its retrospective design, which, combined with the rarity of vulvar cancer, causes the data to be distributed over a period of twelve years. This can be seen in the uneven distribution of radical vulvectomies and SLNBs/IFLs between the US and EBVS groups due to a paradigm

shift in vulvar cancer surgery during the study period [6,39,40]. The more extensive surgeries in the US group were, however, accounted for in the regression analyses, and an independent association between the surgical device and postoperative complications was still observed. Interestingly, the extent of groin procedures was not a significant predictor of postoperative complications in this study; this is likely due to small sample sizes, especially in the "no groin procedures" -subgroup (n = 6), and the heterogeneity of procedures within the IFL group (unilateral IFL, IFL with a contralateral SLNB, or bilateral IFL – all with or without preceding SLNB).

Another important limitation is the absence of systematic frailty screening, which could modify the risk for postoperative complications. However, it is to be noted that there was no statistically significant difference in the patients' age, BMI, or ASA classification between the two groups. Additionally, by including only patients treated with curative intent, the study excluded severely frail patients undergoing palliative surgery.

Among the strengths of this study is the high quality and completeness of the data register used. The use of a standardized, objective primary outcome measure reduces ambiguity and enhanced the clinical relevance of the findings. Furthermore, multivariable regression analyses were performed to control for confounding factors and increase the validity of the findings.

## Implications for practice and future research

A statistically significant difference in postoperative recovery in favor of EBVS devices is clinically relevant, as practices regarding instrument selection in vulvar cancer surgery may not be systematic and uniform. However, the low incidence of vulvar cancer highlights the need for prospective multi-center studies to ensure the generalizability of the results. Important areas for future research include cost-effectiveness analyses for each surgical device and data on the prevalence of long-term complications. Especially the prevalence of lower limb lymphedema, an outcome that may significantly impact both quality of life and health care costs, should be compared in patients with vulvar cancer who were operated on EBVS and US devices considering the devices' unique properties regarding tissue fusion and thermal spread.

## Conclusion

The data from this study suggests electrothermal bipolar vessel sealing devices could reduce early postoperative complications in vulvar cancer surgery compared to ultrasonic devices. Prospective multi-center studies are needed to ensure the generalizability of the results, as the rarity of vulvar cancer is the major limiting factor in achieving sufficient sample size.

## Acknowledgments

None.

## Author contributions

**Conceptualization:** Milla K. Mörsky, Ilkka S. Kaartinen, Reita H. Nyberg.

**Data curation:** Milla K. Mörsky.

**Formal analysis:** Milla K. Mörsky.

**Funding acquisition:** Ilkka S. Kaartinen.

**Investigation:** Milla K. Mörsky, Ilkka S. Kaartinen, Reita H. Nyberg.

**Methodology:** Reita H. Nyberg.

**Supervision:** Ilkka S. Kaartinen, Reita H. Nyberg.

**Writing – original draft:** Milla K. Mörsky.

**Writing – review & editing:** Ilkka S. Kaartinen, Reita H. Nyberg.

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
