## [Decision Letter · Decision Letter 0]

11 Sep 2025

Dear Dr. Mörsky,

Thank you for submitting your manuscript to PLOS ONE. After careful consideration, we feel that it has merit but does not fully meet PLOS ONE’s publication criteria as it currently stands. Therefore, we invite you to submit a revised version of the manuscript that addresses the points raised during the review process.

We look forward to receiving your revised manuscript.

Kind regards,

Eiji Kondo

Academic Editor

PLOS ONE

Journal Requirements:

3. Thank you for uploading your study's underlying data set. Unfortunately, the repository you have noted in your Data Availability statement does not qualify as an acceptable data repository according to PLOS's standards.

Additional Editor Comments:

Reviewer #1:

This article presents a retrospective analysis comparing EBVS and US, both AHD, in the surgical treatment of vulvar cancer.

The manuscript is well structured: the inclusion and exclusion criteria are clearly defined, and both strengths and limitations are appropriately acknowledged. The English language is clear and precise.

The statistical analysis is thoroughly described, with all variables included in the multivariate model properly reported. The Results section is logically organized according to primary and secondary outcomes, which makes the findings easy to follow and interpret. The Discussion is well developed, with comprehensive comparisons to the existing literature, and the Conclusion is concise and to the point.

One point to consider: in the Methods section it is stated that flap reconstruction, when required, was performed by a plastic surgeon. However, this aspect is not further addressed in the manuscript. I would recommend including additional details on flap reconstructions, specifying whether they occurred in either group (EBVS vs. US) and clarifying if they may represent an independent factor for complications or if they are related to the device used.

Reviewer #2:

This is great study to evaluate energy device for vulvar surgery.

I want to confirm one point.

・L327: Does "Clavien-Dindo II" mean "II-V" in Table3? Is this right or error? Please confirm.

The following confirms the fundamental points addressed in the manuscript as a whole.

・"Thermal damage" in US is tissue damage due to abnormal heat of active blade. If so, it might be improved by the surgeon's skill, as other surgery of open or laparoscopic surgery. What do you think about this?

・"Reconstruction" is important step in vulvar dissection, and this procedure is included in both arms to some extent. In this procedure, are energy device used? By gynecologist or cosmetic surgeon? The difference in each arm is thought to be actually important to the outcome. What do you think?

Reviewers' comments:

Reviewer's Responses to Questions

**Comments to the Author**

1. Is the manuscript technically sound, and do the data support the conclusions?

Reviewer #1: Yes

Reviewer #2: Yes

2. Has the statistical analysis been performed appropriately and rigorously?

Reviewer #1: Yes

Reviewer #2: Yes

3. Have the authors made all data underlying the findings in their manuscript fully available?

Reviewer #1: No

Reviewer #2: Yes

4. Is the manuscript presented in an intelligible fashion and written in standard English?

Reviewer #1: Yes

Reviewer #2: Yes

Reviewer #1: This article presents a retrospective analysis comparing EBVS and US, both AHD, in the surgical treatment of vulvar cancer.

The manuscript is well structured: the inclusion and exclusion criteria are clearly defined, and both strengths and limitations are appropriately acknowledged. The English language is clear and precise.

The statistical analysis is thoroughly described, with all variables included in the multivariate model properly reported. The Results section is logically organized according to primary and secondary outcomes, which makes the findings easy to follow and interpret. The Discussion is well developed, with comprehensive comparisons to the existing literature, and the Conclusion is concise and to the point.

One point to consider: in the Methods section it is stated that flap reconstruction, when required, was performed by a plastic surgeon. However, this aspect is not further addressed in the manuscript. I would recommend including additional details on flap reconstructions, specifying whether they occurred in either group (EBVS vs. US) and clarifying if they may represent an independent factor for complications or if they are related to the device used.

Reviewer #2: This is great study to evaluate energy device for vulvar surgery.

I want to confirm one point.

・L327: Does "Clavien-Dindo II" mean "II-V" in Table3? Is this right or error? Please confirm.

The following confirms the fundamental points addressed in the manuscript as a whole.

・"Thermal damage" in US is tissue damage due to abnormal heat of active blade. If so, it might be improved by the surgeon's skill, as other surgery of open or laparoscopic surgery. What do you think about this?

・"Reconstruction" is important step in vulvar dissection, and this procedure is included in both arms to some extent. In this procedure, are energy device used? By gynecologist or cosmetic surgeon? The difference in each arm is thought to be actually important to the outcome. What do you think?

**Do you want your identity to be public for this peer review?** For information about this choice, including consent withdrawal, please see our Privacy Policy

Reviewer #1: No

Reviewer #2: **Yes: ** Fuminori Ito

---

## [Author Response · Author response to Decision Letter 1]

1 Oct 2025

->Thank you. The formatting of the article is now updated to meet the aforementioned standards.

Affiliations updated

Font and headings updated

Captions and legends in figures&tables updated

->The section “Funding information” of the manuscript was updated to match the criteria for Financial Disclosure.

Under “Funding” (L425–435):

Changed “This research was in part supported by grants from the Finnish Society for Research in Gynecology & Obstetrics, the Finnish Society for Gynecological Surgery, Tampere University, and the Competitive State Research Financing of the Expert Responsibility area of the Tampere University Hospital (grant 9AB108).”

To

“This research was in part supported by grants from the Finnish Society for Research in Gynecology & Obstetrics (MM), the Finnish Society for Gynecological Surgery (MM), Tampere University (MM), and the Competitive State Research Financing of the Expert Responsibility area of the Tampere University Hospital (grant 9AB108) (IK). Funder websites: the Finnish Society for Research in Gynecology & Obstetrics - https://gynekologiyhdistys.fi/synnytys-ja-naistentautiopin-tutkimussaatio/; the Finnish Society for Gynecological Surgery - https://www.gks.fi; Tampere University - https://www.tuni.fi/; the Competitive State Research Financing of the Expert Responsibility area of the Tampere University Hospital - https://www.pirha.fi/ammattilaiselle/tutkimus/tutkimusrahoitus/valtion-tutkimusrahoitus. The funders had no role in study design, data collection and analysis, decision to publish, or preparation of the manuscript.”

3. Thank you for uploading your study's underlying data set. Unfortunately, the repository you have noted in your Data Availability statement does not qualify as an acceptable data repository according to PLOS's standards.

->Thank you for this comment. Unfortunately, data sharing is prohibited under current Finnish legislation governing the use of health and social data (the Act on the Secondary Use of Health and Social Data). However, data permits for the same data set can be applied for from Findata. We have now explained this more clearly in the revised version of the manuscript (chapter “Ethical considerations”, and “Data availability statement”) and provided URLs to Findata’s site, where other parties may apply for data permits for the same data set.

under “Ethical considerations” (L218–219): added the sentence “Due to legal restrictions supporting data is not available (see “Data availability statement”).”

Under “Data availability statement” (L437–444):

changed “The results are presented in accordance with the Finnish Social and Health Data Permit Authority Findata’s guidelines on result anonymity; due to legal restrictions supporting data is not available.”

To

“The results are presented in accordance with the Finnish Social and Health Data Permit Authority Findata’s guidelines on result anonymity.

The secondary use of social and health data in Finland is governed by the Act on the Secondary Use of Health and Social Data. Data disclosed under the provisions of this Act may only be accessed by individuals with a research permit and processed within a secure processing environment. Thus, data cannot be shared publicly. Data permits for the same data set can be applied for from Findata (https://findata.fi/en).”

N/A

->Reference list checked. No references were added or removed. DOI:s for references #6 and #35 were corrected.

Additional Editor Comments:

Reviewer #1:

This article presents a retrospective analysis comparing EBVS and US, both AHD, in the surgical treatment of vulvar cancer.

The manuscript is well structured: the inclusion and exclusion criteria are clearly defined, and both strengths and limitations are appropriately acknowledged. The English language is clear and precise.

The statistical analysis is thoroughly described, with all variables included in the multivariate model properly reported. The Results section is logically organized according to primary and secondary outcomes, which makes the findings easy to follow and interpret. The Discussion is well developed, with comprehensive comparisons to the existing literature, and the Conclusion is concise and to the point.

6: One point to consider: in the Methods section it is stated that flap reconstruction, when required, was performed by a plastic surgeon. However, this aspect is not further addressed in the manuscript. I would recommend including additional details on flap reconstructions, specifying whether they occurred in either group (EBVS vs. US) and clarifying if they may represent an independent factor for complications or if they are related to the device used.

->Thank you for this comment. Details on the different types of reconstrutive flaps were added to chapter “Surgical techniques”. The number of reconstructions in each group is presented in Table 2.

“Reconstruction” is included as an independent variable in the multivariable regression models of CD, blood loss and prolonged hospital stay, and it did not have an independently significant effect on these outcomes. As all flaps were local fasciocutaneous flaps, and only 1 vs 2 flaps in the US and EBVS groups were not VY advancement flaps, we find it justifiable handle reconstructions as a single group. Thus, reconstruction does not seem to explain the differences in postoperative complications in this study.

We agree on the importance on this study question, and are working on another manuscript that further explores the relationship between vulvar reconstruction and postoperative complications.

Under “Surgical techniques” (L164–167): changed “All reconstructions were either advancement, propeller, or transposition type fasciocutaneous flaps and were always performed by a plastic surgeon.”

To

“All reconstructions were performed by a plastic surgeon. In all cases, a local fasciocutaneous flap was chosen [VY advancement flap 13/14 (92.9%) vs 20/22 (90.9%) in the US and EBVS groups, respectively; propeller flap 1/14 (7.1%) vs 1/22 (4.5%); transposition flap 0/14 (0.0%) vs 1/22 (4.5%)]. The flaps were mobilized with monopolar diathermy. “

Reviewer #2:

This is great study to evaluate energy device for vulvar surgery.

7: I want to confirm one point.

・L327: Does "Clavien-Dindo II" mean "II-V" in Table3? Is this right or error? Please confirm.

->Thank you for this comment. This is not an error – although we defined “complications” in this study as C-D II – V, the incidence of C-D III and IV is low in both groups, and there are no grade V complications in either group. Thus, the majority of all complications (II – V) are in fact C-D II complications. We rewrote this sentence to clarify this point.

Under “Discussion” (L313–317): changed “In this study, the main driver behind the observed difference in postoperative complications was a lower number of Clavien-Dindo II complications in the EBVS group. This group of complications consisted primarily of infections: postoperative blood transfusions were rare in both groups, and no cases of parenteral nutrition were observed. “

To

“In this study, Clavien-Dindo II was the most frequently occurring class of complications: 70.7% vs 48.9% of all patients in the US and EBVS groups, respectively, were classified as Clavien-Dindo II. This group of complications consisted primarily of infections: postoperative blood transfusions were rare in both groups (9.8% vs ≤6.7%; p=0.19), and no cases of parenteral nutrition were observed.”

8: The following confirms the fundamental points addressed in the manuscript as a whole.

・"Thermal damage" in US is tissue damage due to abnormal heat of active blade. If so, it might be improved by the surgeon's skill, as other surgery of open or laparoscopic surgery. What do you think about this?

->Thank you for an important comment. Surgeon’s skill plays a crucial part in avoiding complications in electrosurgery - with US devices, it is important that the surgeon knows to avoid partial tissue bites and extended activations. However, it might be difficult for any surgeon to avoid relatively fast-paced, repeated activation cycles in vulvar cancer surgery, as there is a large area of tissue to be resected, a low need for dissection and easy access to the surgical field. As there is a substantial increase in the cooling time of US devices when the blade has overheated, surgeons might underestimate the time needed for the device to cool down even when trying to adhere to the manufacturer’s instructions regarding cooling times. In laparoscopy, the slower maneuvering of the device and higher need for dissection might provide more time for the device to spontaneously cool down.

We have added a sentence to the chapter “Discussion” to better convey this.

Under “Discussion” (L332–337):

changed “This could pose a risk for thermal injury when using an US device in vulvar cancer surgery, as multiple consecutive tissue bites may be needed when performing resections of large surface areas; furthermore, the more volatile thermal profile of the US device could make it harder for the surgeon to estimate the cutting blade has reached a safe temperature.”

to

“This could pose a risk for thermal injury when using an US device in vulvar cancer surgery - multiple consecutive tissue bites may be needed when performing resections of large surface areas, and the low need for dissection and ease of maneuvering in an open-surgery setting provide few spontaneous pauses between activation cycles that would facilitate the cool-down of the device. Furthermore, the more volatile thermal profile of the US device could make it harder for the surgeon to estimate whether the cutting blade has reached a safe temperature.

9: ・"Reconstruction" is important step in vulvar dissection, and this procedure is included in both arms to some extent. In this procedure, are energy device used? By gynecologist or cosmetic surgeon? The difference in each arm is thought to be actually important to the outcome. What do you think?

->Thank you for this comment. In our center, reconstructions are always performed by a plastic surgeon. Flaps are raised using monopolar diathermy. Reconstructions seemed to be slightly more common in the EBVS group; however, this difference was not statistically significant (Table 2). The reconstructive techniques in the US and EBVS groups were similar, and are now described in detail in the chapter “Surgical techniques”.

“Reconstruction” was included as an independent variable in the multivariable regression models of CD, blood loss and prolonged hospital stay, and it did not have an independently significant effect on these outcomes. Thus, reconstruction does not seem to explain the differences in postoperative complications in this study. Please see also the answer to question #6.

Changes to the chapter “Surgical techniques”: see answer to question #6

---

## [Decision Letter · Decision Letter 1]

8 Oct 2025

Electrothermal bipolar vessel sealing devices are associated with lower rates of postoperative complications compared to ultrasonic devices in vulvar cancer surgery

PONE-D-25-37760R1

Dear Dr. Mrs Milla Mörsky,

We’re pleased to inform you that your manuscript has been judged scientifically suitable for publication and will be formally accepted for publication once it meets all outstanding technical requirements.

Kind regards,

Eiji Kondo

Academic Editor

PLOS ONE

Additional Editor Comments (optional):

Dear Dr. Mrs Milla Mörsky

Thank you very much for submitting this interesting manuscript.

"Electrothermal bipolar vessel sealing devices are associated with lower rates of postoperative complications compared to ultrasonic devices in vulvar cancer surgery."

We deeply appreciate your support of this journal.

Reviewers' comments:

Reviewer's Responses to Questions

**Comments to the Author**

Reviewer #1: All comments have been addressed

Reviewer #2: All comments have been addressed

2. Is the manuscript technically sound, and do the data support the conclusions?

Reviewer #1: Yes

Reviewer #2: Yes

3. Has the statistical analysis been performed appropriately and rigorously?

Reviewer #1: Yes

Reviewer #2: Yes

4. Have the authors made all data underlying the findings in their manuscript fully available?

Reviewer #1: No

Reviewer #2: Yes

5. Is the manuscript presented in an intelligible fashion and written in standard English?

Reviewer #1: Yes

Reviewer #2: Yes

Reviewer #1: (No Response)

Reviewer #2: The paper was great.

All indications were corrected appropriately.

Thanks for reviewing your great paper.

**Do you want your identity to be public for this peer review?** For information about this choice, including consent withdrawal, please see our Privacy Policy

Reviewer #1: No

Reviewer #2: No

---

## [Editor Report · Acceptance letter]

PONE-D-25-37760R1

PLOS ONE

Dear Dr. Mörsky,

I'm pleased to inform you that your manuscript has been deemed suitable for publication in PLOS ONE. Congratulations! Your manuscript is now being handed over to our production team.

Kind regards,

on behalf of

Dr. Eiji Kondo

Academic Editor

PLOS ONE